# On Learning with a Concurrent Verifier: Convexity, Improving Bounds, and Complex Requirements

## Abstract

Machine learning technologies have been used in a wide range of practical applications. In them, it is preferable to guarantee that the input-output pairs of a model satisfy the given requirements. The recently proposed *concurrent verifier* (CV) is a module combined with machine learning models to guarantee that the model's input-output pairs satisfy the given requirements. The previous paper provides a generalization analysis of learning with a CV to show how the model's learnability changes using a CV. Although the paper provides basic learnability results, many CV properties remain unrevealed. Moreover, the previous work assumed a CV always works correctly, and requirements are imposed on a single input-output pair, which limits the situation where we can use a CV. We show the learning algorithms that preserve convexity when using a CV. We also show conditions that using a CV improves the generalization error bound. Moreover, we analyze the learnability when a CV is incorrect, or requirements are imposed on the combination of multiple input-output pairs.

## 1 Introduction

The recent progress of machine learning technologies enables them to be used in many practical systems. However, when developing a system that uses a machine learning model, a model with small prediction errors might be unsatisfactory since some errors cause severe problems (Amodei et al., 2016) even if they occur with a low probability. Potential problems might surface when we apply machine learning to such security-critical domains as finance, health, and transportation, and we have to pay more on realizing safe systems (Braiek & Khomh, 2020; Xu & Saleh, 2021). Similarly, the recent progress of large language models (Brown et al., 2020; Chowdhery et al., 2022) has improved performance on many NLP tasks, although their undesirable outputs are considered to be important problems (Gehman et al., 2020; Sheng et al., 2021). Therefore, ML models with which we can guarantee that the input-output pairs of a machine learning model satisfy the given *requirements* are highly demanding. We can avoid serious errors if we know that some label $y$ must not be assigned to input $x$ having a specific property and use the knowledge as requirements. Similarly, we can control the output of language models by representing the desired properties as requirements and generating words while satisfying them (Qin et al., 2022; Zhang et al., 2023). Unfortunately, since the models used in modern machine learning tasks tend to be complex, it is unrealistic to train a model to guarantee that every possible input-output pair of a model satisfies requirements. Moreover, some requirements would be unknown when we train a model, which frequently happens when we use pre-trained models.

Learning with a concurrent verifier (CV) (Nishino et al., 2022) is a recently proposed framework that attaches a verifier module to a machine learning model to check the model's input-output pairs and modifies its outputs to guarantee that its input-output pairs satisfy the given requirements. That is, given machine learning model $h : \mathcal{X} \rightarrow \mathcal{Y}$ and requirements $c : \mathcal{X} \times \mathcal{Y} \rightarrow \{0, 1\}$, which is represented as a mapping from an input-output pair to $\{0, 1\}$, a CV checks whether pair $(x, h(x))$ satisfies $c(x, h(x)) = 1$ every time we predict $h(x)$ for input $x \in \mathcal{X}$. If $c(x, h(x)) = 1$, then the CV allows $h(x)$ to be output. If $c(x, h(x)) = 0$, the CV modifies $h(x)$ to another value, $y' \in \mathcal{Y}$, that satisfies $c(x, y') = 1$. Figure 1 shows the overview of the framework of learning with a CV. A CV can be combined with various machine learning models to guarantee that the input-output pairs

of a model satisfy the requirements. Since many task-specific models with low prediction errors are used in practice, adding a CV is reasonable if it does not sacrifice the model's performance.

To analyze the effect of using a CV, a previous paper (Nishino et al., 2022) theoretically analyzed how an upper bound of generalization error changes by adding a CV to a machine learning model. It proved that if a machine learning model is probably approximately correct (PAC) learnable, then using a CV not at the training phase but only at the inference phase guarantees that the model's generalization error is not worse than the other possible models. The paper also showed that the generalization error can be worse than other possible models if not PAC learnable. The paper also showed that if a CV is used in both the training and inference phase, the generalization error bounds based on Rademacher complexity are not worse than the bounds of the original model for any combination of requirement $c$ and hypothesis class $\mathcal{H}$. These theoretical results can give further insights to existing works. For example, existing works (Leino et al., 2022; Qin et al., 2022) impose constraints on an ML model that can be seen as using a verifier in the inference phase but not in the learning phase. The analyses of the paper show there is a potential to improve the generalization error of the model if we know the requirements at a training phase.

Although the previous paper's analyses revealed the basic properties of the problem setting of learning with a CV, it failed to identify many fundamental CV properties, especially efficiency and the potential to improve the bounds. If using a CV makes the learning problem inefficient, then its usefulness might be limited. Although requirements are not always used to improve prediction accuracy, it is useful to know when we can improve generalization error bounds by using a CV since using background knowledge Moreover, the previous paper restricts the form of requirements $c(x, y)$. First, it assumes that a verifier can correctly evaluate $c(x, h(x))$ for any pair $(x, h(x))$. This assumption might not be appropriate since evaluating $c(x, y)$ is a hard problem for some requirements. Second, it assumes the requirements are posed on a single input-output pair $(x, y)$. However, a machine learning model is sometimes needed to satisfy the requirements of multiple input-output pairs, i.e., we want to deal with requirements of form $c(x_1, h(x_1), \ldots, x_n, h(x_n))$, where $(x_i, h(x_i))$ is an input-output pair for model $h$. We show motivating examples for these types of requirements in Sec.1.1.

This paper addresses the above problems to reveal the important properties of a CV. Our main findings are as follows:

**Preserving convexity:** A CV preserves convexity on typical multi-class convex learning problems, including logistic regression, multi-class SVM, and multi-class AdaBoost. We further generalize the results to convex loss functions having a specific form (§5).

**Bound improvements:** We can reduce the Rademacher complexity by using a CV if the requirements uniquely determine possible labels for input $x$ (§6).

**Incorrect verifiers:** We show that a generalization bound will not change if we use an incorrect CV instead of a correct one. Moreover, we can give statistical assurance to an incorrect CV by accessing annotated samples (§7).

**Verifying requirements over multiple samples:** If requirements are posed on multiple input-output pairs, we can obtain the generalization bounds similar to those for the original CV. If a model is PAC-learnable, we can obtain a bound even if we use a CV only at inference time. If we use a CV both in the training and inference phases, we can obtain generalization error bounds based on the Rademacher complexity of the original model (§8).

## 1.1 USE CASES OF CV

Nishino et al. (2022) shows some use cases of CV. We add a few more use cases, especially for the new extensions introduced in this paper.

**Safety-critical domains:** To guarantee that input-output pairs of a model satisfy requirements is important for safety-critical domains. For example, if we use a classification model for deciding suitable therapy $y$ for patient $x$, there are inadvisable choices for patients with specific properties, so-called contraindications. Using a CV, we can guarantee that the model does not violate known contraindications. Moreover, Leino et al. (2022) proposes an ML model for an airborne collision avoidance system whose output satisfies safety constraints.

**Incorrect verifiers:** There are cases where we cannot access perfect verifiers. For example, if we want to prohibit a text summarization model from generating a summary inconsistent with the input, judging consistency is difficult (Kryscinski et al., 2020). Moreover, there are cases where the

knowledge base is incomplete, or it might be computationally expensive to handle a complete set of rules in a CV.

**Requirements over multiple samples:** We need requirements over multiple input-output pairs to guarantee consistency among outputs. Suppose we want to assign jobs to $n$ workers $x_1, \ldots, x_n$ using model $h$ that maps worker $x$ to job $y$. Using a classification model that maps a worker to a job, we can decide assignments independently as $h(x_1), \ldots, h(x_n)$. However, such assignments might be unsatisfactory since some jobs will not be assigned to any worker. Requirement over multiple samples can help this problem. Another typical example is requirements on robustness. Given inputs $x_1, \ldots, x_n$, and prediction over them $h(x_1) \ldots, h(x_n)$, we say $h$ is robust with small changes if $h(x_i) = h(x_j)$ if $x_i$ and $x_j$ are close enough.

## 2    RELATED WORK

Machine learning models that can exploit constraints have been investigated in many research fields. For example, Markov logic networks (Richardson & Domingos, 2006), Problogs (De Raedt et al., 2007), and probabilistic circuit models (Kisa et al., 2014; Poon & Domingos, 2011) integrate statistical models with symbolic logic formulations. Since these models can incorporate hard constraints represented by symbolic logic, they can guarantee input-output pairs. More recent approaches can be found in the survey paper on using logical constraints with deep neural networks (Giunchiglia et al., 2022). For structured prediction, many approaches have been considered to control the language generation process to satisfy soft and hard constraints (Qin et al., 2022; Zhang et al., 2023). Previous research focused on their practical performance and gave little theoretical analysis of their learnability when hard constraints are used.

The verification of machine learning models continues to gather attention. Attempts have verified whether a machine learning model can provide specific desired properties (Bunel et al., 2018; Tjeng et al., 2019; Katz et al., 2017; Singh et al., 2018). Exact verification methods use integer programming (MIP) (Tjeng et al., 2019), constraint satisfaction (SAT) (Narodytska et al., 2020), and a satisfiable module theory (SMT) solver (Katz et al., 2017) to assess the robustness of a neural network model against input noise. These approaches aim to obtain models that fulfill the required properties. However, verification methods cannot help modify the models if they do not satisfy such requirements. If we want ML models to meet specific requirements, post-processing is needed as a concurrent verification model.

Other lines of approaches combine an additional module with a ML model to realize "concurrent verification". Semantic probabilistic layer (SPL) (Ahmed et al., 2022) puts an additional layer implemented using a probabilistic circuit to a neural network to guarantee that input-output pairs satisfy requirements. Runtime shielding (Zhu et al., 2019) avoids unsafe explorations in reinforcement learning by prohibiting unsafe actions. SC-Net (Leino et al., 2022) changes outputs of neural networks to satisfy ordering constraints. Although they are closely related to a CV, no generalization analyses are given to these models. Since SPL and SC-Net can be seen as adding a verifier to the existing model, theoretical results on CV can be applied to these models.

Various methods can give upper bounds on generalization errors, including VC-dimension (Vapnik & Chervonenkis, 1971) and its extensions (Daniely et al., 2015; Natarajan, 1989), Rademacher complexity (Bartlett & Mendelson, 2003; Koltchinskii & Panchenko, 2002), stability (Shalev-Shwartz et al., 2010), PAC-Bayes (McAllester, 1998; Alquier, 2021), and information-theoretic bounds (Zhang, 2006; Xu & Raginsky, 2017). Following (Nishino et al., 2022), we use Rademacher complexity in the following analysis since it is among the most popular tools for giving theoretical upper bounds on generalization error.

## 3    PRELIMINARIES

Let $\mathcal{X}$ denote the domain of the inputs, and let $\mathcal{Y}$ be the domain of the labels. Let $\mathcal{H}$ be a hypothesis class, a set of measurable functions $h : \mathcal{X} \to \mathcal{Y}$. Training sample $S = ((x_1, y_1), \ldots, (x_m, y_m)) \in (\mathcal{X} \times \mathcal{Y})^m$ is a finite sequence of size $m$ drawn i.i.d. from a fixed but unknown probability distribution $\mathcal{D}$ on $\mathcal{X} \times \mathcal{Y}$. We represent set $\{1, \ldots, K\}$ as $[K]$. In the following sections, we assume $\mathcal{Y} = [K]$ unless otherwise stated. Given distribution $\mathcal{D}$ on $\mathcal{X} \times \mathcal{Y}$, $\hat{L}_{\mathcal{D}}(h)$ denotes *the generalization*

*error* and $L_S(h)$ denotes *the empirical error* of $h$ over $S$:

$$L_{\mathcal{D}}(h) := \mathop{\mathbb{E}}_{(x,y)\sim\mathcal{D}}[\mathbf{1}_{h(x)\neq y}], \quad \hat{L}_S(h) := \frac{1}{m}\sum_{i=1}^{m}\mathbf{1}_{h(x_i)\neq y_i}, \tag{1}$$

where $\mathbf{1}_\omega$ is the indicator function of event $\omega$. Let $A$ be a learning algorithm that maps sample $S$ to hypothesis $h \in \mathcal{H}$.

**PAC learnability:** We say hypothesis class $\mathcal{H}$ is *agnostic PAC-learnable* if there exists function $m_{\mathcal{H}} : (0,1)^2 \to \mathbb{N}$ and learning algorithm $A$ with the following property: For every $\epsilon, \delta \in (0,1)$ and distribution $\mathcal{D}$ over $\mathcal{X} \times \mathcal{Y}$, if $S$ consists of $m \geq m_{\mathcal{H}}(\epsilon, \delta)$ i.i.d. examples generated by $\mathcal{D}$, then with at least probability $1 - \delta$, the following holds:

$$L_{\mathcal{D}}(A(S)) \leq \min_{h\in\mathcal{H}} L_{\mathcal{D}}(h) + \epsilon. \tag{2}$$

Distribution $\mathcal{D}$ is *realizable* by hypothesis set $\mathcal{H}$ if $h^* \in \mathcal{H}$ exists such that $L_{\mathcal{D}}(h^*) = 0$. If $\mathcal{D}$ is realizable by agnostic PAC-learnable hypothesis $\mathcal{H}$, then $\mathcal{H}$ is *PAC-learnable*. If $\mathcal{H}$ is PAC-learnable, then (2) becomes $L_{\mathcal{D}}(A(S)) \leq \epsilon$ since $\min_{h\in\mathcal{H}} L_{\mathcal{D}}(h) = 0$.

**Rademacher complexity:** Let $\ell : \mathcal{Y} \times \mathcal{Y} \to \mathbb{R}$ be a loss function. Given loss function $\ell$ and hypothesis class $\mathcal{H}$, we define $\mathcal{G} := \ell \circ \mathcal{H} := \{(x,y) \mapsto \ell(h(x), y) : h \in \mathcal{H}\}$. We use Rademacher complexity to derive the generalization bounds.

**Definition 3.1.** (Empirical Rademacher complexity) Let $\mathcal{G}$ be a family of functions mapping from $\mathcal{X} \times \mathcal{Y}$ to $\mathbb{R}$, and let $S = ((x_1, y_1), \ldots, (x_m, y_m)) \in (\mathcal{X} \times \mathcal{Y})^m$ be a sample of size $m$. Then *the empirical Rademacher complexity* of $\mathcal{G}$ with respect to $S$ is defined as

$$\hat{\mathcal{R}}_S(\mathcal{G}) := \mathop{\mathbb{E}}_{\boldsymbol{\sigma}}\left[\sup_{g\in\mathcal{G}}\sum_{i=1}^{m}\sigma_i g(x_i, y_i)\right],$$

where $\boldsymbol{\sigma} = (\sigma_1, \ldots, \sigma_m) \in \{\pm 1\}^m$ are random variables distributed i.i.d. according to $\mathbb{P}[\sigma_i = 1] = \mathbb{P}[\sigma_i = -1] = 1/2$, which are called Rademacher variables. For any $m \geq 1$, *the Rademacher complexity* of $\mathcal{G}$ is defined as the expectation of the empirical Rademacher complexity over all the samples of size $m$ drawn based on $\mathcal{D}$ as $\mathcal{R}_m(\mathcal{G}) := \mathbb{E}_{S\sim\mathcal{D}^m}[\hat{\mathcal{R}}_S(\mathcal{G})]$.

## 4 CONCURRENT VERIFIER

We review the definition of CV and the main results of Nishino et al. (2022). CV works with a machine learning model corresponding to a measurable mapping $h : \mathcal{X} \to \mathcal{Y}$. If $x$ is given to the model, which outputs $h(x)$, then the CV checks whether the pair $(x, h(x))$ satisfies the requirements. We assume that the requirements can be represented as *requirement function* $c : (\mathcal{X} \times \mathcal{Y}) \to \{0, 1\}$. If $c(x, h(x)) = 1$, then the pair satisfies the requirement; if $c(x, h(x)) = 0$, then it does not. For simplicity, we assume that for every possible input $x \in \mathcal{X}$, there exists $y \in \mathcal{Y}$ such that $c(x, y) = 1$.

After checking the input-output pair, a CV modifies output $h(x)$ depending on the value of $c(x, h(x))$. If $c(x, h(x)) = 1$, the CV outputs $h(x)$ since it satisfies the requirements. If $c(x, h(x)) = 0$, then it modifies $h(x)$ to some $y \in \mathcal{Y}$ that satisfies $c(x, y) = 1$. If we use a CV with a machine learning model that corresponds to $h$, then the combination of the model and the CV can be seen as a function $h_c : \mathcal{X} \to \mathcal{Y}$:

$$h_c(x) := \begin{cases} h(x) & \text{if } c(x, h(x)) = 1 \\ y_c & \text{if } c(x, h(x)) = 0 \end{cases}, \tag{3}$$

where $y_c \in \mathcal{Y}$ satisfies $c(x, y_c) = 1$ and is selected deterministically for every pair of $h, x$. When $\mathcal{Y} = [K]$, an example of selecting minimum integer $i \in [K]$ satisfying $c(x, i) = 1$ as $y_c$ is a reasonable choice. When $\mathcal{Y} = [K]$ and $h(x)$ is made by scoring functions $h(x, y) : (\mathcal{X} \times \mathcal{Y}) \to \mathbb{R}$, it is also reasonable to select $y^*$ such that $y^* = \text{argmax}_{y\in\mathcal{Y}, c(x,y)=1} h(x, y)$.

In the following analysis of the multi-class classification setting, we assume that every hypothesis is defined based on scoring function $h : \mathcal{X} \times \mathcal{Y} \to \mathbb{R}$, which defines mapping $x \mapsto \text{argmax}_{y\in\mathcal{Y}} h(x, y)$. Using a CV that corresponds to modifying scoring function $h(x, y)$ to $h_c(x, y)$, which is defined as

$$h_c(x, y) := \begin{cases} h(x, y) & \text{if } c(x, h(x)) = 1 \\ C & \text{if } c(x, h(x)) = 0 \end{cases}, \tag{4}$$

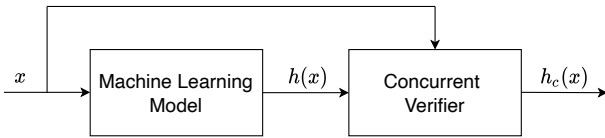

Figure 1: Overview of a machine learning model with a CV that checks whether input-output pairs of a model satisfy requirements.

where $C$ is a constant that satisfies $C < \inf_{x,y} h(x, y)$. Fig. 1 shows an overview of using a CV with a machine learning model.

Learning a model corresponds to selecting hypothesis $h$ from hypothesis class $\mathcal{H}$. Therefore, learning a model with a CV corresponds to choosing a hypothesis from the modified hypothesis class: $\mathcal{H}_c = \{h_c : h \in \mathcal{H}\}$. Since every hypothesis in $\mathcal{H}_c$ satisfies the requirements, we can guarantee that the model satisfies the condition if we select a hypothesis from $\mathcal{H}_c$. In the following sections, we analyze the learnability of $\mathcal{H}_c$ by comparing it with that of $\mathcal{H}$.

Next, we review some CV properties. Nishino et al. (2022) analyzed generalization errors in two different situations depending on when to use a CV. The first setting is the *inference time verification* (ITV). Since we do not use a CV when we learn model $\hat{h} \in \mathcal{H}$ from training sample $S$, we add a CV to $\hat{h}$ to make a combined model $\hat{h}_c$ to perform inference. This setting corresponds to a situation where the requirements are unknown when we train the model. The second setting is *learning time verification* (LTV), where we use a CV in both the learning and inference phases. In other words, we select hypothesis $\hat{h}_c \in \mathcal{H}_c$ in the learning phase using a training sample and use $\hat{h}_c$ in the inference phase. ITV is a more flexible setting than LTV since we do not need to know the requirements in the training phase for it. On the other hand, we expect to find a better hypothesis in the LTV setting since we can consider the effect of the CV on selecting $\hat{h}_c$.

We get the following result for the ITV setting.

**Theorem 4.1** (Theorem 5.1 of (Nishino et al., 2022)). *If $\mathcal{Y} = [K]$, and hypothesis class $\mathcal{H}$ is PAC-learnable with training sample $S$, then suppose that $\hat{h} \in \mathcal{H}$ is a hypothesis estimated from $S$ satisfying $L_{\mathcal{D}}(\hat{h}) \leq \epsilon$ for parameter $\epsilon \in (0, 1)$. Then, for any requirement $c$, hypothesis $\hat{h}_c$ obtained by modifying $\hat{h}$ with a CV satisfies*

$$L_{\mathcal{D}}(\hat{h}_c) \leq \min_{h_c \in \mathcal{H}_c} L_{\mathcal{D}}(h_c) + \epsilon \,.$$

This theorem shows that if $\mathcal{H}$ is PAC-learnable, using a CV only at the inference phase is sufficient to obtain a model with a small generalization error among $\mathcal{H}_c$. The previous paper also showed that if $\mathcal{H}$ is not PAC-learnable, then $L_{\mathcal{D}}(\hat{h}_c)$ can be much larger than the other hypothesis in $\mathcal{H}_c$.

On the LTV setting, the following bound holds:

**Theorem 4.2** (Theorem 6.1 of (Nishino et al., 2022)). *Let $\mathcal{H} \subseteq \mathbb{R}^{\mathcal{X} \times \mathcal{Y}}$ be a hypothesis class with $\mathcal{Y} = [K]$, and let $c$ be a requirement. Fix $\rho > 0$. Then for any $\delta > 0$, with probability at least $1 - \delta$, the following bound holds for every $h_c \in \mathcal{H}_c$:*

$$L_{\mathcal{D}}(h_c) \leq \hat{L}_{S,\rho}(h_c) + \frac{4K}{\rho}\mathcal{R}_m(\Pi_1(\mathcal{H})) + \sqrt{\frac{\log \frac{1}{\delta}}{2m}} \,, \tag{5}$$

*where $\Pi_1(\mathcal{H})$ is defined as $\Pi_1(\mathcal{H}) := \{x \mapsto h(x, y) : y \in \mathcal{Y}, h \in \mathcal{H}\}$.*

$L_{S,\rho}(h)$ is an empirical margin loss defined as $\hat{L}_{S,\rho}(h) := \frac{1}{m} \sum_{i=1}^{m} \Phi_\rho(\rho_h(x_i, y_i))$, where $\rho_h(x, y)$ is *the margin of function* of $h$ defined as $\rho_h(x, y) := h(x, y) - \max_{y' \neq y} h(x, y')$, and $\Phi_\rho(t)$ is the margin loss defined as $\Phi_\rho(t) := \min(1, \max(0, 1 - t/\rho))$. By comparing the above bound with the standard Rademacher complexity-based bound for multi-class classification (e.g., Theorem 9.2 of (Mohri et al., 2012)), we can see that the second and the third terms of the RHS of (5) equal to that of the standard one. It means using a CV during a learning phase does not worsen the error bound for any requirement $c$.

## 5 PRESERVING CONVEXITY

We analyzed the effect of using a CV on efficiency in multi-class classification. Using a CV requires additional computation in both the learning and inference phases. In the latter, we have to evaluate $c(x, y)$ for $y \in [K]$ for given input $x$ to compute the scoring function $h_c(x, y)$. Therefore, the additional computation cost equals $K$ evaluations of $c(x, y)$. In the learning phase, we must evaluate $c(x_i, y)$ for every $x_i \in S$ and $y \in [K]$. Moreover, modifying $h(x, y)$ to $h_c(x, y)$ might change the relationship between the objective function and the parameters, complicating the optimization problem. In this section, we assume that the set of hypothesis $\mathcal{H}$ is represented as a real-valued parameter vector $\mathbf{w} \in \mathbb{R}^d$. For many machine learning models, a learning problem can be formulated as an optimization problem for finding $\mathbf{w} \in \Theta \subseteq \mathbb{R}^d$ that minimizes an objective function.

If $\Theta$ is a convex set and the objective function is convex regarding $\mathbf{w}$, then the optimization problem is convex. A convex optimization problem is an important class of such tasks since finding optimal solutions is easy. For some machine learning models, including logistic regression, support vector machines (SVMs), and AdaBoost, their learning problems correspond to convex optimization problems. We show that using a CV on these models does not damage the convexity of learning problems.

**Multi-class Logistic Regression**  Multi-class logistic regression is a probabilistic model that defines conditional distribution $\mathbb{P}[y = k \mid x] = \exp(a_k) / \sum_{j=1}^{K} \exp(a_j)$, where $a_k = \mathbf{w}_k^\mathrm{T} \phi(x)$ and we assume $(\mathbf{w}_1, \ldots, \mathbf{w}_K) \in \mathbb{R}^{K \times d}$ and $\phi$ is a feature mapping $\phi : \mathcal{X} \to \mathbb{R}^d$. The cross-entropy error of the logistic regression model is a convex function of parameters $(\mathbf{w}_1, \ldots, \mathbf{w}_K)$ (e.g., (Bishop, 2006)).

As stated in the previous section, using a CV for a multi-class classification problem corresponds to modifying scoring functions $h(x, y)$ to $h_c(x, y)$. Since multi-class logistic regression models make decisions based on score function $h(x, k) = a_k$, the modified model defines the conditional distribution

$$\mathbb{P}[y = k \mid x] = \frac{\exp(b_k)}{\sum_{j=1}^{K} \exp(b_j)}, \tag{6}$$

by substituting $h(x, k)$ with $h_c(x, k) = b_k$, where $b_k = a_k$ if $c(x, k) = 1$, otherwise $b_k = C < \inf_{x, \mathbf{w}_k} \mathbf{w}_k^\mathrm{T} \phi(x)$. We show the cross-entropy loss of the above modified logistic regression model is convex.

**Theorem 5.1.** *The optimization problem of minimizing the cross-entropy loss of the multi-class logistic regression model combined with a CV with regard to $(\mathbf{w}_1, \ldots, \mathbf{w}_K) \in \mathbb{R}^{K \times d}$ is a convex optimization problem.*

We give the proof in Appendix A. The proof shows the semi-definiteness of the hessian. Intuitively, using a CV corresponds to deleting class labels with $c(x, k) = 0$ since $h(x, k)$ is a small constant. It makes the original problem $K'$-class classification problem where $K' < K$. Since the learning problem of logistic regression is convex for any $K \geq 1$, a CV preserves convexity.

**Multi-class SVM**  Multi-class SVM (Crammer & Singer, 2000; 2002) is an extension of SVM for multi-class classification. Similar to logistic regression, a multi-class SVM learns a decision function that is linear with parameters $\mathbf{W} = (\mathbf{w}_1, \ldots, \mathbf{w}_K)$ and has form $x \mapsto \mathrm{argmax}_{k \in K} \mathbf{w}_k^\mathrm{T} \phi(x)$. Estimating parameters $\mathbf{W}$ is represented as the following optimization problem:

$$\min_{\mathbf{W}, \boldsymbol{\xi}} \frac{1}{2} \sum_{k=1}^{K} \|\mathbf{w}_k\|^2 + B \sum_{i=1}^{m} \xi_i$$

$$\text{subject to: } \forall i \in [m], \forall k \in [K] - \{y_i\}, \xi_i \geq 0, \mathbf{w}_{y_i}^\mathrm{T} \phi(x_i) \geq \mathbf{w}_k^\mathrm{T} \phi(x_i) + 1 - \xi_i.$$

Since the objective is convex with $\mathbf{W}$ and $\boldsymbol{\xi}$ and the constraints are affine, the above problem is a convex optimization problem. Since the constraints are affine and differentiable, there exists an equivalent dual problem which is also a convex optimization problem. We usually solve it to learn the parameters.

Applying a CV to a multi-class SVM corresponds to changing the decision function to $x \mapsto \arg\max_{k \in [K]} h_c(x, k)$, where $h_c(x, k) = \mathbf{w}_k^T \boldsymbol{\phi}(x)$ if $c(x, k) = 1$; otherwise, $h_c(x, k)$ is a small constant. Using a CV corresponds to changing the first constraint of the optimization problem. We can remove the constraint for $c(x_i, l) = 0$ since inequality is always satisfied for such cases. Note that we assume $c(x_i, y_i) = 1$ for all $i \in [m]$ since the model will always fail to classify the sample with $c(x_i, y_i) = 0$, and removing such samples will not affect the learning procedure.

This modified optimization problem is a convex optimization problem since the objective is convex with respect to $\mathbf{W}$ and $\boldsymbol{\xi}$, and the constraints are affine. Moreover, since the problem satisfies weak Slater's condition (Mohri et al., 2012), a strong duality holds, and the corresponding dual optimization problem is also convex.

**Multi-class AdaBoost** AdaBoost.MR (Schapire & Singer, 1999; 2000) is a multi-class extension of AdaBoost (Freund & Schapire, 1997). Let $\mathcal{H}_b = \{h_1, \ldots, h_n\}$ be a family of base hypothesis mapping $\mathcal{X} \times \mathcal{Y}$ to $\{-1, +1\}$. The learning problem of AdaBoost.MR solves the minimization problem of objective function $F$ defined for samples $S = ((x_1, y_1), \ldots, (x_m, y_m)) \in (\mathcal{X} \times \mathcal{Y})^M$ and $\bar{\boldsymbol{\alpha}} = (\bar{\alpha}_1, \ldots, \bar{\alpha}_n) \in \mathbb{R}^n$, $n \geq 1$, by

$$F(\bar{\boldsymbol{\alpha}}) = \sum_{i=1}^m \sum_{k \neq y_i} e^{-\sum_{j=1}^n \bar{\alpha}_j (h_j(x_i, y_i) - h_j(x_i, k))} .$$

$F$ is convex since it is a sum of convex functions, each obtained by the composition of the exponential function with an affine function of $\bar{\boldsymbol{\alpha}}$. $F$ is also differentiable since the exponential function is differentiable.

AdaBoost.MR defines the scoring function $h(x, k)$ as $\sum_{j=1}^n \bar{\alpha}_j h_j(x, k)$. Adding a CV to AdaBoost.MR corresponds to substituting $h(x, y)$ with $h_c(x, y)$, which preserves the convexity of $F$ since it is still a sum of the compositions of exponential functions and an affine function of $\bar{\boldsymbol{\alpha}}$. Similarly, $F$ remains differentiable when using a CV.

The above observation on AdaBoost.MR leads us to a more general case by using standard composition rules of convex functions (Boyd & Vandenberghe, 2004). Let $h_k = h(x, k)$. If we fix $x$, then $h_k$ can be seen as a mapping from $\mathbf{w}$ to $\mathbb{R}$. Let $g : \mathbb{R}^K \to \mathbb{R}$. If the loss for pair $(x, y)$ is defined as a composition of the form $g(h_1, \ldots, h_k)$, we can say the loss is convex if (i) $g$ is convex and non-decreasing in each argument, and $h_k$ is convex, or (ii) $g$ is convex and non-increasing in each argument, and $h_k$ is concave. If the loss of a model satisfies either of the above conditions, then using a CV preserves convexity since using a CV corresponds to substituting some $h_k$ with $c(x, k) = 0$ to a constant value, and such substitution does not break the above conditions on function compositions.

## 6 IMPROVING GENERALIZATION BOUNDS

Theorem 4.2 shows that using a CV in the LTV setting does not increase the bound on generalization error. This result is positive since we can use any requirements without heeding the increase of the generalization error. On the other hand, it remains unclear whether using a CV improves the generalization error bound. Whether we can improve the bound depends on the type of requirements. Therefore, we consider a bound that depends on requirement functions $c$ and $\mathcal{D}$.

We say requirement function $c$ *uniquely determines* label of $x$ to $\hat{y}$ if $c(x, y) = 0$ for all $y \neq \hat{y}$ and $c(x, \hat{y}) = 1$. We assume that score function $h_c(x, \hat{y}) = M$ for all $h_c \in \mathcal{H}_c$ if $c$ uniquely determines the label of $x$, where $M$ is a constant satisfying $C \ll M$. This assumption is safe since it does not change the predictions made by each hypothesis $h_c \in \mathcal{H}_c$. Let $\mathcal{D}_{\mathcal{X}}$ be the marginal distribution over $\mathcal{X}$, and $\mathcal{C}_0 \subseteq \mathcal{X}$ be the set of $x$ where $c$ uniquely determines labels of $x$. Let $p_0 = \mathbb{E}_{x \sim \mathcal{D}_{\mathcal{X}}}[\mathbf{1}_{x \in \mathcal{C}_0}]$, i.e., the probability that $x \in \mathcal{C}_0$ is generated from $\mathcal{D}_{\mathcal{X}}$. Since $p_0$ is the probability that the CV uniquely determines the label of input $x$, we can bound the Rademacher complexity of $\mathcal{H}_c$ by using $p_0$. Let $\mathcal{D}_0$ and $\mathcal{D}_1$ be the conditional distribution of $x$ conditioned on the event $x \in \mathcal{C}_0$ and $x \notin \mathcal{C}_0$, respectively. Then, $\mathcal{D}_{\mathcal{X}}$ can be written as $\mathcal{D}_{\mathcal{X}} = p_0 \mathcal{D}_0 + (1 - p_0)\mathcal{D}_1$. Using $p_0$, we can obtain the following bound of the Rademacher complexity.

**Theorem 6.1.** *Let $\mathcal{H} \subseteq \mathbb{R}^{\mathcal{X} \times \mathcal{Y}}$ be a hypothesis class with $\mathcal{Y} = [K]$, and let $c$ be a requirement. Suppose that every pair $(x, y) \in \mathcal{X} \times \mathcal{Y}$ is generated by sample distribution $\mathcal{D}$ and $p_0 = \mathbb{E}_{x \sim \mathcal{D}_{\mathcal{X}}}[\mathbf{1}_{x \in \mathcal{C}_0}]$.*

*Fix $\rho > 0$. Then for any $\delta > 0$ and $m > 0$ with probability at least $1 - \delta$, the following bound holds for every $h_c \in \mathcal{H}_c$:*

$$L_{\mathcal{D}}(h_c) \leq \hat{L}_{S,\rho}(h_c) + \frac{4K}{\rho} \sum_{k=0}^{m-1} \mathfrak{B}[k; m, p_0] \frac{m-k}{m} \mathcal{R}_{m-k}^{\mathcal{D}_1}(\Pi_1(\mathcal{H})) + \sqrt{\frac{\log \frac{1}{\delta}}{2m}}, \qquad (7)$$

*where $\mathcal{R}_{m-k}^{\mathcal{D}_1}(\Pi_1(\mathcal{H}))$ is the Rademacher complexity of $\Pi_1(\mathcal{H})$ computed on distribution $\mathcal{D}_1$, and $\mathfrak{B}[k; m, p_0]$ represents the Binomial distribution $\mathfrak{B}[k; m, p_0] := \binom{m}{k} p_0^k (1-p_0)^{m-k}$.*

We show the full proof in Appendix B. The proof uses the fact that all $h_c(x, y)$ have the same value for $x \in \mathcal{C}_0$ to show that the empirical Rademacher complexity of samples generated from $\mathcal{D}_0$ is zero. Compared with Theorem 4.2, the theorem shows that the Rademacher complexity of the modified hypothesis can be bounded by the weighted sum of the Rademacher complexities of $\mathcal{H}$ defined with sample size $k = 1, \ldots, m$ and distribution $\mathcal{D}_1$. This inequality will not directly improve the bound since the Rademacher complexity depends on $\mathcal{D}_1$, which differs from the original $\mathcal{D}_{\mathcal{X}}$. However, since there are known upper bounds on the Rademacher complexity that do not depend on the form of data distribution, combining these distribution agnostic bounds will result in improved upper bounds. For some models, $\mathcal{R}_m(\Pi_1(\mathcal{H}))$ has an upper bound $\mathcal{O}(1/\sqrt{m})$, which does not depend on $\mathcal{D}$ (Mohri et al., 2012). If we can write $\mathcal{R}_m(\Pi_1(\mathcal{H})) \leq D/\sqrt{m}$ with constant $D$, then we can show that the second term of the RHS of (7) is strictly smaller than $4KD/\rho\sqrt{m}$, an upper bound of $\frac{4K}{\rho} \mathcal{R}_m(\Pi_1(\mathcal{H}))$, if $p_0 > 0$. We can have tighter bounds if requirement $c$ determines more labels on elements $x \sim \mathcal{D}_{\mathcal{X}}$. Note that $p_0$ is unclear in a practical setting since $\mathcal{D}$ is unknown. However, we can estimate $p_0$ from samples drawn from $\mathcal{D}_{\mathcal{X}}$ by evaluating whether $x \in \mathcal{C}_0$ or not.

## 7 INCORRECT VERIFIER

Nishino et al. (2022) assumes we can always access the true requirement function $c$. This assumption is reasonable for many requirements. For example, if $y$ requires a sentence containing specific words or phrases, then the requirement evaluation is trivial. On the other hand, as shown in the use cases of Sec. 1.1, judging whether the input-output pair satisfies a requirement is sometimes complicated. Let $c$ be a true requirement function that we cannot directly access or whose accessing is highly expensive; instead, we access incorrect function $c' : (\mathcal{X} \times \mathcal{Y}) \to \{0, 1\}$. Thus, we consider a setting where a CV uses $c'$ instead of $c$. We can analyze the effect of using $c'$ from two different aspects. The first is its effect on the generalization error on hypothesis set $\mathcal{H}$. Since the effect of using a CV on the generalization error only reflects how we modify each hypothesis with a CV, whether requirement functions $c'$ are correct does not affect the generalization ability. Therefore, we can apply generalization bounds for a correct $c$ to cases where $c'$ might be incorrect.

The second aspect concerns what kind of guarantees we can provide for the hypotheses modified with $c'$. We must measure the deviation between $c$ and $c'$ to evaluate this case. Since the deviation between them depends on data generation distribution $\mathcal{D}$ and selected hypothesis $\mathcal{H}$, we resort to a task-specific approach, a straightforward approach that assesses the deviation using true $c$ to annotate the outputs of $h_c$. We first learn hypothesis $\hat{h}_c \in \mathcal{H}_c$ that minimizes the training error, dip into $N$ samples from $\mathcal{D}_{\mathcal{X}}$, and use $\hat{h}_c$ to predict its labels. Next, we access $c$ to annotate pairs $(x, \hat{h}_c(x))$ to evaluate the probability that $\hat{h}_c$ satisfies the requirement. Since whether the obtained results satisfy the requirement follows a Binomial distribution, we can obtain an upper bound of the error ratio of $c'$ by applying a standard technique to derive tail bounds of Binomial distributions.

## 8 VERIFYING REQUIREMENTS OVER MULTIPLE SAMPLES

Our analysis so far assumes that a requirement on a model is always imposed independently on a single input-output pair $(x, y) \in \mathcal{X} \times \mathcal{Y}$. However, as we mentioned in the introduction, we want to deal with requirements that deal with multiple pairs. Let $\mathcal{H}$ be the set of hypothesis $h : \mathcal{X} \to \mathcal{Y}$, where $\mathcal{Y} = [K]$. Let $h^n : \mathcal{X}^n \to \mathcal{Y}^n$ be the hypothesis $h$ extended for applications to multiple inputs, i.e., $h^n(\mathbf{x}) = (h(x_1), \ldots, h(x_n)) \in \mathcal{Y}^n$. Let $c : (\mathcal{X} \times \mathcal{Y})^n \to \{0, 1\}$ be the requirement function over $n$ pairs. Using a CV corresponds to modifying hypothesis $h^n$ to $h_c^n$, where $h_c^n(\mathbf{x})$ equals $h^n(\mathbf{x})$ if $c(\mathbf{x}, h^n(\mathbf{x})) = 1$; otherwise it returns $\mathbf{y}' \in \mathcal{Y}^n$ satisfying $c(\mathbf{x}, \mathbf{y}') = 1$. Below, we

use different rules for selecting $\mathbf{y}'$. The first rule selects $\mathbf{y}'$ that satisfies $c(\mathbf{x}, \mathbf{y}') = 1$ and minimizes the Hamming distance from $h^n(\mathbf{x})$. The second rule assumes that $h$ decides $\mathbf{y}$ based on a scoring function, and we select $\mathbf{y}'$ satisfying $c(\mathbf{x}, \mathbf{y}') = 1$ and having the maximum score. Let $\mathsf{L}(h^n(\mathbf{x}), \mathbf{y})$ be the sum of 0-1 loss defined as $\mathsf{L}(h^n(\mathbf{x}), \mathbf{y}) := \frac{1}{n} \sum_{i=1}^{n} \mathbf{1}_{h(x_i) \neq y_i}$. If no requirement is given, the loss coincides with the average of 0-1 losses for i.i.d samples.

As with a setting with single input-output pairs, we consider the ITV and LTV settings. First, we can obtain a generalization error bound in the ITV setting if $\mathcal{H}$ is PAC-learnable.

**Theorem 8.1.** *Suppose that $\mathcal{H}$ is realizable and PAC-learnable. If $\hat{h} \in \mathcal{H}$ satisfies $L_{\mathcal{D}}(h) \leq \epsilon$, then for any requirement $c$ on $n$ tuples, we have the following bound:*

$$\mathbb{E}_{(\mathbf{x},\mathbf{y})\sim\mathcal{D}^n}[\mathsf{L}(\hat{h}_c^n(\mathbf{x}), \mathbf{y})] \leq \min_{h^c \in \mathcal{H}_c} \mathbb{E}_{(\mathbf{x},\mathbf{y})\sim\mathcal{D}^n}[\mathsf{L}(h_c^n(\mathbf{x}), \mathbf{y})] + n\epsilon. \tag{8}$$

We show the full proof in Appendix C. The proof derives the bound by comparing the generalization error of $\hat{h}_c^n$ with the best hypothesis $f_c^n$, and shows the difference between them can be bounded by $n\epsilon$ if we use the above rule for selecting $\mathbf{y}'$ based on the minimum Hamming distance.

We extend the Rademacher complexity-based bound on the LTV settings to the multiple samples setting. In the following, we assume that hypothesis $h$ is a mapping defined by scoring function $h(x, y)$, and $h^n(\mathbf{x}) = \mathrm{argmax}_{\mathbf{y}} h^n(\mathbf{x}, \mathbf{y})$, where we define the score function as $h^n(\mathbf{x}, \mathbf{y}) := \frac{1}{n} \sum_{i=1}^{n} h(x_i, y_i)$. In this setting, using a CV corresponds to modifying score function $h^n(\mathbf{x}, \mathbf{y})$ to $h_c^n(\mathbf{x}, \mathbf{y})$ defined as

$$h_c^n(\mathbf{x}, \mathbf{y}) := \begin{cases} h^n(\mathbf{x}, \mathbf{y}) & \text{if } c(\mathbf{x}, \mathbf{y}) = 1 \\ C & \text{if } c(\mathbf{x}, \mathbf{y}) = 0 \end{cases}, \tag{9}$$

where $C < \inf_{\mathbf{x},\mathbf{y}} h^n(\mathbf{x}, \mathbf{y})$. Let $\rho_{h^n}(\mathbf{x}, \mathbf{y})$ be a margin function defined as

$$\rho_{h^n}(\mathbf{x}, \mathbf{y}) := h^n(\mathbf{x}, \mathbf{y}) - \max_{\mathbf{y}' \neq \mathbf{y}} h^n(\mathbf{x}, \mathbf{y}').$$

We have the following generalization bound based on Rademacher complexity. Here, we define a single example as a tuple of $n$ pairs $(x, y) \in \mathcal{X} \times \mathcal{Y}$ sampled from $\mathcal{D}^n$. Let $S$ be a set of $M$ such samples.

**Theorem 8.2.** *Let $\mathcal{H} \subseteq \mathbb{R}^{\mathcal{X} \times \mathcal{Y}}$ be a hypothesis class with $\mathcal{Y} = [K]$, $S = ((x_{11}, y_{11}), \ldots ((x_{1n}, y_{1n})), \ldots, ((x_{m1}, y_{m1}), \ldots, (x_{mn}, y_{mn}))$, and let $c$ be a requirement. Fix $\rho > 0$. Then for any $\delta > 0$, with probability at least $1 - \delta$, the following bound holds for all $h_c \in \mathcal{H}_c$:*

$$\mathbb{E}_{(\mathbf{x},\mathbf{y})\sim\mathcal{D}^n}[\mathsf{L}(\hat{h}_c^n(\mathbf{x}), \mathbf{y})] \leq L_{S,\rho}(h_c^n) + \frac{4K\sqrt{2n}}{\rho}\mathcal{R}_{mn}(\Pi_1(\mathcal{H})) + \sqrt{\frac{\log\frac{1}{\delta}}{2m}}. \tag{10}$$

We give the proof in Appendix D. The core of the proof is giving a bound to the Rademacher complexity of the modified hypothesis by the Rademacher complexity of $\mathcal{H}$. We use an extension of Taragrand's lemma introduced by Cortes et al. (2016) to derive the bound. Comparing the above bound with the bound for a single input-output case (Theorem 4.2), we see that the second term of the RHS changes from $\frac{4K}{\rho}\mathcal{R}_m(\Pi_1(\mathcal{H}))$ to $\frac{4K\sqrt{2n}}{\rho}\mathcal{R}_{mn}(\Pi_1(\mathcal{H}))$. Since many upper bounds of the Rademacher complexity scales in $\mathcal{O}(1/\sqrt{m})$, the order of these bounds would be compatible. Since we use $m \times n$ pairs in the case of multiple pairs, we need $n$ times more data to obtain the compatible bound in this case. The last term of RHS is the same as the single input-output pair case. But we must note that we use $n$ times more samples since we use $n$-tuples for training.

## 9 CONCLUSION

This paper analyzed the fundamental properties of learning with a concurrent verifier. Combined with the theoretical results of the previous paper, we can provide a comprehensive understanding of learning with a CV. A recent survey paper (Giunchiglia et al., 2022) on deep neural networks with constraints envisions that giving certification will become essential for machine learning applications. Since adding a verifier to an ML model is a simple solution, our theoretical analyses would contribute to this line of work.

## REPRODUCIBILITY STATEMENTS

We give complete proofs for all the theoretical claims of the paper in the appendix.

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

## A   PROOF OF THEOREM 5.1

*Proof.* Since using a CV does not change the domain of feasible values of $\mathbf{w}_1, \ldots, \mathbf{w}_K$, we show that the cross-entropy loss function remains a convex function. Let $\mathbf{t}_i \in \{0,1\}^K$ be a one-hot vector representing the label of the $i$-th training sample. Given training sample $S = ((x_1, y_1), \ldots, (x_m, y_m))$, the cross-entropy loss $E(\mathbf{w}_1, \ldots, \mathbf{w}_K)$ is defined as

$$E(\mathbf{w}_1, \ldots, \mathbf{w}_K) := -\sum_{i=1}^{m} \sum_{j=1}^{K} t_{ij} \ln \mathbb{P}[y = j \mid x_i].$$

The derivative of $E$ with respect to $\mathbf{w}_j$ is

$$\frac{\partial E(\mathbf{W})}{\partial \mathbf{w}_j} = \sum_{i=1}^{m} \sum_{k=1}^{K} \frac{\partial E}{\partial y_{ik}} \frac{\partial y_{ik}}{\partial b_{ij}} \frac{\partial b_{ij}}{\partial a_{ij}} \frac{\partial a_{ij}}{\partial \mathbf{w}_j}$$

$$= -\sum_{i=1}^{m} \sum_{k=1}^{K} \frac{t_{ik}}{y_{ik}} y_{ik} (\delta_{kj} - y_{ij}) c_{ij} \boldsymbol{\phi}_i$$

$$= \sum_{i=1}^{m} (y_{ij} - t_{ij}) c_{ij} \boldsymbol{\phi}_i,$$

where $y_{ik}$ is $\mathbb{P}[y = k \mid x_i]$, $\boldsymbol{\phi}_i = \phi(x_i)$ and $\delta_{jk} = 1$ if $j = k$, otherwise $\delta_{jk} = 0$. $c_{ik} = \partial b_{ik}/\partial a_{ik}$, and $c_{ik} = 1$ if $c(x_i, k) = 1$, otherwise $c_{ik} = 0$. The hessian $\mathbf{H}$ is an $Kd \times Kd$ matrix whose

$(k, j)$-th submatrix of size $d \times d$ is

$$\frac{\partial^2 E}{\partial \mathbf{w}_k \partial \mathbf{w}_j} = \sum_{i=1}^{m} \frac{\partial y_{ik}}{\partial \mathbf{w}_j} c_{ik} \boldsymbol{\phi}_i$$

$$= \sum_{i=1}^{m} \frac{\partial y_{ik}}{\partial b_{ij}} \frac{\partial b_{ij}}{\partial a_{ij}} \frac{\partial a_{ij}}{\partial \mathbf{w}_j} c_{ik} \boldsymbol{\phi}_i$$

$$= \sum_{i=1}^{m} y_{ik}(\delta_{kj} - y_{ij}) c_{ik} c_{ij} \boldsymbol{\phi}_i \boldsymbol{\phi}_i^{\mathrm{T}} .$$

We prove the convexity by showing that $\mathbf{u}^{\mathrm{T}} \mathbf{H} \mathbf{u} \geq 0$ for any $\mathbf{u} \in \mathbb{R}^{Kd}$.

$$\mathbf{u}^{\mathrm{T}} \mathbf{H} \mathbf{u} = \sum_{i=1}^{m} \sum_{j,k} \alpha_{ij} y_{ik}(\delta_{kj} - y_j) c_{ij} c_{ik} \alpha_{ik}$$

$$= \sum_{i=1}^{m} \sum_{k} y_{ik} c_{ik}^2 \alpha_{ik}^2 - \left( \sum_{k} y_{ik} c_{ik} \alpha_{ik} \right)^2 , \tag{11}$$

where $\alpha_{ik} = \sum_{j=1}^{d} u_{kj} \phi_{ij}$. By using the fact $\sum_k y_{ik} = 1$ and $f(x) = x^2$ is a convex function, we can apply Jensen's inequality to get the following inequality for all $i \in [m]$:

$$\sum_{k} y_{ik} c_{ik}^2 \alpha_{ik}^2 = \sum_{k} y_{ik} f(c_{ik} \alpha_{ik}) \geq f\left( \sum_{k} y_{ik} c_{ik} \alpha_{ik} \right) = \left( \sum_{k} y_{ik} c_{ik} \alpha_{ik} \right)^2 .$$

Thus we can see $\mathbf{u}^{\mathrm{T}} \mathbf{H} \mathbf{u} \geq 0$. $\qquad \square$

## B  PROOF OF THEOREM 6.1

*Proof.* The proof first follows the proof for Theorem 9.2 of Mohri et al. (2012). We first define hypothesis class $\mathcal{H}_0 := \{(x, y) \mapsto \rho_{\theta, h_c}(x, y) : h_c \in \mathcal{H}_c\}$, where $\rho_{\theta, h} := \min_{y'}(h(x, y) - h(x, y') + \theta \mathbf{1}_{y'=y})$. We fix $\theta = 2\rho$. Following the proof for Theorem 9.2 of Mohri et al. (2012), we have the following bound for any $h \in \mathcal{H}_c$ and $\delta > 0$ with probability at least $1 - \delta$:

$$L_{\mathcal{D}}(h_c) \leq \hat{L}_{S, \rho}(h_c) + \frac{2}{\rho} \mathcal{R}_m(\mathcal{H}_0) + \sqrt{\frac{\log \frac{1}{\delta}}{2m}} . \tag{12}$$

In the following, we show that $\mathcal{R}_m(\mathcal{H}_0) \leq 2K \sum_{k=0}^{m-1} \mathfrak{B}[k; m, p_0] \frac{m-k}{m} \mathcal{R}_{m-k}^{\mathcal{D}_1}(\Pi_1(\mathcal{H}))$.

$\mathcal{R}_m(\mathcal{H}_0)$ can be upper-bounded as follows:

$$\mathcal{R}_m(\mathcal{H}_0) = \frac{1}{m} \mathop{\mathbb{E}}_{S, \sigma} \left[ \sup_{h_c \in \mathcal{H}_c} \sum_{i=1}^{m} \sigma_i(h_c(x_i, y_i) - \max_{y}(h_c(x_i, y) - 2\rho \mathbf{1}_{y=y_i})) \right]$$

$$\leq \frac{1}{m} \mathop{\mathbb{E}}_{S, \sigma} \left[ \sup_{h_c \in \mathcal{H}_c} \sum_{i=1}^{m} \sigma_i h_c(x_i, y_i) \right] + \frac{1}{m} \mathop{\mathbb{E}}_{S, \sigma} \left[ \sup_{h_c \in \mathcal{H}_c} \sum_{i=1}^{m} \sigma_i \max_{y}(h_c(x_i, y) - 2\rho \mathbf{1}_{y=y_i}) \right] . \tag{13}$$

Now, we bound the first term above. Suppose training sample $S$ is divided into $S_0$ and $S_1$, where $S_0$ is a set of samples $x \in \mathcal{C}_0$, and $S_1$ is a set of samples $x \notin \mathcal{C}_0$, respectively. Let $|S_0| = k < m$. Since the empirical Rademacher complexity is invariant of the order of $x_i$, we suppose that $x_1, \dots, x_k \in S_0$ and $x_{k+1}, \dots, x_m \in S_1$. Then, empirical Rademacher complexity corresponding to the first term of (13) can be bounded as

$$\frac{1}{m} \mathop{\mathbb{E}}_{\sigma} \left[ \sup_{h_c \in \mathcal{H}_c} \sum_{i=1}^{m} \sigma_i h_c(x_i, y_i) \right] = \frac{1}{m} \mathop{\mathbb{E}}_{\sigma} \left[ \sup_{h_c \in \mathcal{H}_c} \sum_{i=1}^{k} \sigma_i h_c(x_i, y_i) + \sum_{i=k+1}^{m} \sigma_i h_c(x_i, y_i) \right]$$

$$\leq \frac{1}{m} \mathop{\mathbb{E}}_{\sigma} \left[ \sup_{h_c \in \mathcal{H}_c} \sum_{i=1}^{k} \sigma_i h_c(x_i, y_i) \right] + \frac{1}{m} \mathop{\mathbb{E}}_{\sigma} \left[ \sup_{h_c \in \mathcal{H}_c} \sum_{i=k+1}^{m} \sigma_i h_c(x_i, y_i) \right] ,$$

where we use the sub-additivity of sup. From the assumption that $c$ uniquely determines labels for $x_i \in S_0$, the first term above equals to zero since $h_c(x_i, y_i)$ takes the same value for all $h_c \in \mathcal{H}_c$, and the empirical Rademacher complexity of the singleton hypothesis is zero. Following the proof of Theorem 6.1 of Nishino et al. (2022), we have that the second term above is bounded as

$$\frac{1}{m} \mathbb{E}_{\sigma} \left[ \sup_{h_c \in \mathcal{H}_c} \sum_{i=k+1}^{m} \sigma_i h_c(x_i, y_i) \right] \leq \frac{m-k}{m} K \mathcal{R}_{S_1}(\Pi_1(\mathcal{H})). \tag{14}$$

Similarly, we bound the empirical Rademacher complexity corresponding to the second term of (13) as:

$$\frac{1}{m} \mathbb{E}_{\sigma} \left[ \sup_{h_c \in \mathcal{H}_c} \sum_{i=1}^{m} \sigma_i \max_y (h_c(x_i, y) - 2\rho \mathbf{1}_{y=y_i}) \right] \leq \frac{1}{m} \mathbb{E}_{S,\sigma} \left[ \sup_{h_c \in \mathcal{H}_c} \sum_{i=1}^{k} \sigma_i \max_y (h_c(x_i, y) - 2\rho \mathbf{1}_{y=y_i}) \right]$$
$$+ \frac{1}{m} \mathbb{E}_{\sigma} \left[ \sup_{h_c \in \mathcal{H}_c} \sum_{i=k+1}^{m} \sigma_i \max_y (h_c(x_i, y) - 2\rho \mathbf{1}_{y=y_i}) \right].$$

The first term of the RHS of the above equals zero since all $h_c in \mathcal{H}_c$ takes the same value for $x_i \in \mathcal{C}_0$. Following the proof of Theorem 6.1 of Nishino et al. (2022), the second term above is bounded as

$$\frac{1}{m} \mathbb{E}_{\sigma} \left[ \sup_{h_c \in \mathcal{H}_c} \sum_{i=k+1}^{m} \sigma_i \max_y (h_c(x_i, y) - 2\rho \mathbf{1}_{y=y_i}) \right] \leq \frac{m-k}{m} K \mathcal{R}_{S_1}(\Pi_1(\mathcal{H})). \tag{15}$$

Combining (14) and (15), we have the following bound for the empirical Rademacher complexity $\mathcal{R}_{(S_0,S_1)}(\mathcal{H}_0)$:

$$\mathcal{R}_{(S_0,S_1)}(\mathcal{H}_0) \leq \frac{m-k}{m} 2K \mathcal{R}_{S_1}(\Pi_1(\mathcal{H})).$$

Next, we derive a bound of $\mathcal{R}_m(\mathcal{H}_0)$. Since the Rademacher complexity $\mathcal{R}_m(\mathcal{H}_0)$ is the expectation of empirical Rademacher complexity, we have:

$$\mathcal{R}_m(\mathcal{H}_0) = \mathbb{E}_{S=(x_1,\ldots,x_m)\sim\mathcal{D}_{\mathcal{X}}^m} [\mathcal{R}_S(\mathcal{H}_0)]$$
$$= \sum_{k=0}^{m} \mathfrak{B}[k; m, p_0] \mathbb{E}_{(S_0,S_1)\sim\mathcal{D}_0^k \times \mathcal{D}_1^{m-k}} [\mathcal{R}_{(S_0,S_1)}(\mathcal{H}_0)]$$
$$\leq \sum_{k=0}^{m} \mathfrak{B}[k; m, p_0] \frac{m-k}{m} 2K \mathcal{R}_{m-k}(\Pi_1(\mathcal{H})),$$

which completes the proof. $\qquad\square$

## C    PROOF OF THEOREM 8.1

*Proof.* Since $\mathcal{H}$ is realizable, there exists $f \in \mathcal{H}$ such that $L_{\mathcal{D}}(f) = 0$. From the definition, $f$ also satisfies $\mathbb{E}_{(x,y)^n \sim \mathcal{D}^n}[L(f^n(\mathbf{x}), \mathbf{y})] = 0$. Similarly, $\mathbb{E}_{(x,y)^n \sim \mathcal{D}^n}[L(\hat{h}^n(\mathbf{x}), \mathbf{y})] \leq 1/n \sum_{i=1}^{n} \mathbb{E}(x,y) \sim \mathcal{D}[\mathbf{1}_{h(x_i)\neq y_i}] \leq \epsilon$ from the definition of L. Let $f_c^n$ be the mapping that satisfies requirements and achieves the minimum error. We prove the theorem by bounding $\mathbb{E}_{(x,y)^n \sim \mathcal{D}^n}[L(h_c^n(\mathbf{x}), \mathbf{y}) - L(f_c^n(\mathbf{x}), \mathbf{y})]$. When pair $(\mathbf{x}, \mathbf{y})$ satisfies $L(\hat{h}^n(\mathbf{x}), \mathbf{y}) = 0$, then $L(h_c^n(\mathbf{x}), \mathbf{y}) - L(f_c^n(\mathbf{x}), \mathbf{y}) = 0$ since the minimum hamming distance correction rule can select $\mathbf{y}'$ with minimum errors for both $\hat{h}_c^n$ and $f_c^n$ if $c(\mathbf{x}, \mathbf{y}) = 0$. If $L(\hat{h}^n(\mathbf{x}), \mathbf{y}) > 0$, then $L(\hat{h}_c^n(\mathbf{x}), \mathbf{y}) - L(f_c^n(\mathbf{x}), \mathbf{y})$ can be at most $n$ times larger than $L(h_c^n(\mathbf{x}), \mathbf{y}) - L(f_c^n(\mathbf{x}), \mathbf{y})$ since the modified predictions can be incorrect for all $y_i$. Therefore, for any $(x,y)^n \sim \mathcal{D}^n$ the following holds almost surely

$$L(h_c^n(\mathbf{x}), \mathbf{y}) - L(f_c^n(\mathbf{x}), \mathbf{y}) \leq n(L(h^n(\mathbf{x}), \mathbf{y}) - L(f^n(\mathbf{x}), \mathbf{y})).$$

By taking expectation on $(x, y) \sim \mathcal{D}^n$ for the both sides, we have

$$
\begin{aligned}
\mathbb{E}_{(x,y)\sim\mathcal{D}^n}[\mathsf{L}(h_c^n(\mathbf{x}), \mathbf{y})] &\leq \mathbb{E}_{(x,y)^n\sim\mathcal{D}^n}[\mathsf{L}(f_c^n(\mathbf{x}), \mathbf{y})] + n \mathbb{E}_{(x,y)^n\sim\mathcal{D}^n}[\mathsf{L}(\hat{h}^n(\mathbf{x}), \mathbf{y})] \\
&\leq \min_{h_c\in\mathcal{H}_c} \mathbb{E}_{(x,y)^n\sim\mathcal{D}^n}[\mathsf{L}(h_c^n(\mathbf{x}), \mathbf{y})] + n\epsilon \,.
\end{aligned}
$$

$\square$

## D    PROOF OF THEOREM 8.2

To prove the above theorem, we use the following extension of the contraction lemma proved by Cortes et al. (2016).

**Lemma D.1** (Cortes et al. (2016)). *Let $\mathcal{H}$ be a hypothesis set of functions mapping $\mathcal{X}$ to $\mathbb{R}^c$. Assume that for all $i = 1, \dots, m$, $\Psi_i : \mathbb{R}^c \to \mathbb{R}$ is $\mu_i$-Lipschitz for $\mathbb{R}^d$ equipped with the 2-norm. That is:*

$$
|\Psi_i(\mathbf{x}') - \Psi_i(\mathbf{x})| \leq \mu_i \|\mathbf{x}' - \mathbf{x}\|_2 \,,
$$

*for all $(\mathbf{x}, \mathbf{x}') \in (\mathbb{R}^c)^2$. Then, for any sample $S$ of $m$ points $x_1, \dots, x_m \in \mathcal{X}$, the following inequality holds*

$$
\frac{1}{m} \mathbb{E}_{\boldsymbol{\sigma}}\left[ \sup_{\mathbf{h}\in\mathcal{H}} \sum_{i=1}^m \sigma_i \Phi_i(\mathbf{h}(x_i)) \right] \leq \frac{\sqrt{2}}{m} \mathbb{E}_{\boldsymbol{\epsilon}}\left[ \sup_{\mathbf{h}\in\mathcal{H}} \sum_{i=1}^m \sum_{j=1}^c \epsilon_{ij} \mu_i (h_j(x_i)) \right] \,, \tag{16}
$$

*where $\boldsymbol{\epsilon} = (\epsilon_{ij})_{i,j}$ and $\epsilon_{ij}$s are independent Rademacher variables uniformly distributed over $\{\pm 1\}$.*

*Proof.* We first define hypothesis classes $\mathcal{H}_0$ and $\mathcal{H}_1$ as follows:

$$
\begin{aligned}
\mathcal{H}_0 &= \{(\mathbf{x}, \mathbf{y}) \mapsto \rho_{h_c^n}(\mathbf{x}, \mathbf{y}) : h_c^n \in \mathcal{H}_c^n\} \\
\mathcal{H}_1 &= \{\Phi_\rho \circ \hat{h} : \hat{h} \in \mathcal{H}_0\} \,.
\end{aligned}
$$

By applying a technique to derive the error bound based on the Rademacher complexity (e.g., Mohri et al. (2012)) based on the McDiarmid's inequality, we have the following:

$$
\mathbb{E}_{(\mathbf{x},\mathbf{y})\sim\mathcal{D}^n}[\Phi_\rho(\rho_{h_c^n}(\mathbf{x}, \mathbf{y})] \leq \frac{1}{m} \sum_{i=1}^m \Phi_\rho(\rho_{h_c^n}(\mathbf{x}_i, \mathbf{y}_m)) + 2\mathcal{R}_m(\mathcal{H}_1) + \sqrt{\frac{\log\frac{1}{\delta}}{2m}}
$$

Since $\mathbb{E}_{(\mathbf{x},\mathbf{y})\sim\mathcal{D}^n}[\mathsf{L}(h_c^n(\mathbf{x}), \mathbf{y})] \leq \mathbb{E}_{(\mathbf{x},\mathbf{y})\sim\mathcal{D}^n}[\mathbf{1}_{\mathbf{h_c^n}(\mathbf{x})\neq\mathbf{y}}] = \mathbb{E}_{(\mathbf{x},\mathbf{y})\sim\mathcal{D}^n}[\mathbf{1}_{\rho_{\mathbf{h_c^n}}(\mathbf{x},\mathbf{y})\leq\mathbf{0}}] \leq \mathbb{E}_{(\mathbf{x},\mathbf{y})\sim\mathcal{D}^n}[\Phi_\rho(\rho_{\mathbf{h_c^n}}(\mathbf{x}, \mathbf{y}))]$, we have that the RHS of the equation is an upper bound of the generalization error. Since the margin function is $1/\rho$ Lipshitz, we have $\mathcal{R}_m(\mathcal{H}_1) \leq \frac{1}{\rho}\mathcal{R}_m(\mathcal{H}_0)$ from the Talagrand's contraction lemma. Next we upper-bound $\mathcal{R}_m(\mathcal{H}_0)$ as follows:

$$
\begin{aligned}
\mathcal{R}_m(\mathcal{H}_0) &= \frac{1}{m} \mathbb{E}_{S,\boldsymbol{\sigma}}\left[ \sup_{h_c^n\in\mathcal{H}_c^n} \sum_{i=1}^m \sigma_i (h_c^n(\mathbf{x}_i, \mathbf{y}_i) - \max_{\mathbf{y}\neq\mathbf{y}_i}(h_c^n(\mathbf{x}_i, \mathbf{y}))) \right] \\
&\leq \frac{1}{m} \mathbb{E}_{S,\boldsymbol{\sigma}}\left[ \sup_{h_c^n\in\mathcal{H}_c^n} \sum_{i=1}^m \sigma_i h_c^n(\mathbf{x}_i, \mathbf{y}_i) \right] + \frac{1}{m} \mathbb{E}_{S,\boldsymbol{\sigma}}\left[ \sup_{h_c^n\in\mathcal{H}_c^n} \sum_{i=1}^m \sigma_i \max_{\mathbf{y}\neq\mathbf{y}_i} h_c^n(\mathbf{x}_i, \mathbf{y}) \right] \,.
\end{aligned}
$$

Now we bound the second term above. We fist show the Lipschitzness of $h \mapsto \max_{\mathbf{y}' \neq \mathbf{y}} h_c^n(\mathbf{x}, \mathbf{y}')$. Observe that the following chain of inequalities holds for any $h, \tilde{h} \in \mathcal{H}$

$$
\begin{aligned}
& |\max_{\mathbf{y}' \neq \mathbf{y}} h_c^n(\mathbf{x}, \mathbf{y}') - \max_{\mathbf{y}' \neq \mathbf{y}} \tilde{h}_c^n(\mathbf{x}, \mathbf{y}')| \\
& \leq \max_{\mathbf{y}' \neq \mathbf{y}} |h_c^n(\mathbf{x}, \mathbf{y}') - \tilde{h}_c^n(\mathbf{x}, \mathbf{y}')| \\
& \leq \max_{\mathbf{y}} |h_c^n(\mathbf{x}, \mathbf{y}) - \tilde{h}_c^n(\mathbf{x}, \mathbf{y})| \\
& \leq \max_{\mathbf{y}} |h^n(\mathbf{x}, \mathbf{y}) - \tilde{h}^n(\mathbf{x}, \mathbf{y})| \\
& = \frac{1}{n} \max_{\mathbf{y}} |\sum_{i=1}^{n} (h(x_i, y_i) - \tilde{h}(x_i, y_i))| \\
& \leq \frac{1}{n} \sum_{i=1}^{n} \max_{y_i} |h(x_i, y_i) - \tilde{h}(x_i, y_i)| \\
& \leq \frac{\sqrt{n}}{n} \sqrt{\sum_{i=1}^{n} \left[ \max_{y_i} |h(x_i, y_i) - \tilde{h}(x_i, y_i)| \right]^2} \\
& = \frac{1}{\sqrt{n}} \sqrt{\sum_{i=1}^{n} \max_{y_i} |h(x_i, y_i) - \tilde{h}(x_i, y_i)|^2} \\
& \leq \frac{1}{\sqrt{n}} \sqrt{\sum_{i=1}^{n} \sum_{k \in [K]} |h(x_i, k) - \tilde{h}(x_i, k)|^2} \, ,
\end{aligned}
$$

where we use $|h_c^n(\mathbf{x}, \mathbf{y}) - \tilde{h}_c^n(\mathbf{x}, \mathbf{y})| \leq |h^n(\mathbf{x}, \mathbf{y}) - \tilde{h}^n(\mathbf{x}, \mathbf{y})|$ for any $\mathbf{y}$ at line 4. Therefore, by applying lemma D.1, we obtain

$$
\begin{aligned}
& \frac{1}{m} \mathop{\mathbb{E}}_{S, \boldsymbol{\sigma}} \left[ \sup_{h_c^n \in \mathcal{H}_c^n} \sum_{i=1}^{m} \sigma_i \max_{\mathbf{y} \neq \mathbf{y}_i} h_c^n(\mathbf{x}_i, \mathbf{y}) \right] \\
& \leq \frac{\sqrt{2}}{m} \mathop{\mathbb{E}}_{S, \boldsymbol{\epsilon}} \left[ \sup_{h \in \mathcal{H}} \frac{1}{\sqrt{n}} \sum_{i=1}^{m} \sum_{j=1}^{n} \sum_{k=1}^{K} \epsilon_{ijk} h(x_{ij}, k) \right] \\
& \leq \frac{\sqrt{2}}{m} \mathop{\mathbb{E}}_{S, \boldsymbol{\epsilon}} \left[ \sup_{h \in \mathcal{H}} \frac{1}{\sqrt{n}} \sum_{i=1}^{m} \sum_{j=1}^{n} \sum_{k=1}^{K} \epsilon_{ijk} h(x_{ij}, k) \right] \\
& \leq \frac{\sqrt{2}}{m\sqrt{n}} \mathop{\mathbb{E}}_{S, \boldsymbol{\epsilon}} \left[ \sum_k \sup_{h \in \mathcal{H}} \sum_i \sum_j \epsilon_{ijk} h(x_{ij}, k) \right] \\
& \leq \frac{\sqrt{2}}{m\sqrt{n}} \sum_{k \in [K]} \mathop{\mathbb{E}}_{S, \boldsymbol{\epsilon}} \left[ \sup_{h \in \mathcal{H}} \sum_i \sum_j \epsilon_{ijk} h(x_{ij}, k) \right] \\
& \leq \sum_{k \in [K]} \sqrt{2n} \mathcal{R}_{mn}(\Pi_1(\mathcal{H})) \leq K\sqrt{2n} \mathcal{R}_{mn}(\Pi_1(\mathcal{H})) \, ,
\end{aligned}
$$

where we use the sub-additivity of sup at line 4. We also use the fact that $(x_{i1}, \ldots, x_{in})$ are i.i.d. at the last line. We next bound the first term. We have the following Lipschitzness.

$$
\left| h_c(\mathbf{x}_i, \mathbf{y}_i) - \tilde{h}_c(\mathbf{x}_i, \mathbf{y}_i) \right|
$$
$$
\leq \max_{\mathbf{y}} \left| h_c(\mathbf{x}_i, \mathbf{y}) - \tilde{h}_c(\mathbf{x}_i, \mathbf{y}) \right|
$$
$$
\leq \frac{1}{n} \sqrt{n} \sqrt{ \sum_{j=1}^{n} \sum_{k \in [K]} |h(x_{ij}, k) - \tilde{h}(x_{ij}, k)|^2 } \, .
$$

Hence we have

$$
\frac{1}{m} \mathop{\mathbb{E}}_{\boldsymbol{\sigma}} \left[ \sup_{h \in \mathcal{H}_c} \sum_{i=1}^{m} h_c(\mathbf{x}_i, \mathbf{y}_i) \right]
$$
$$
\leq \frac{\sqrt{2}}{m\sqrt{n}} \mathop{\mathbb{E}}_{\boldsymbol{\epsilon}} \left[ \sum_k \sup_{h \in \mathcal{H}} \sum_i \sum_j \epsilon_{ijk} h(x_{ij}, k) \right]
$$
$$
\leq \sum_k \sqrt{2n} \mathcal{R}_{nm}(\Pi_1(\mathcal{H})) \leq K\sqrt{2n} \mathcal{R}_{nm}(\Pi_1(\mathcal{H})) \, .
$$

Hence we can prove $\mathcal{R}_m(\mathcal{H}_1) \leq \frac{2K\sqrt{2n}}{\rho} \mathcal{R}_{nm}(\Pi_1(\mathcal{H}))$. $\qquad\square$

