# OpenReview forum: "On Learning with a Concurrent Verifier: Convexity, Improving Bounds, and Complex Requirements"
_ICLR.cc/2024/Conference — Submitted to ICLR 2024_

### Official Review · Reviewer_A15n · 2023-11-01

**Soundness:** 3 good
**Presentation:** 2 fair
**Contribution:** 3 good
**Rating:** 5
**Confidence:** 3

**Summary:**

Learning with a concurrent verifier is a framework proposed recently where a verified module is coupled with a machine learning model to check the model's input-output pairs and modifies the outputs to guarantee that its input-output pairs satisfy the given requirements. This paper is a follow up of the original paper proposing this framework. More specifically, in this submission, the authors prove that a concurrent verifier preserves convexity on typical multi-class convex learning problems such as logistic regression, multi-class SVM, etc. The authors also improve the generalisation error bound in some conditions, besides removing some restrictions inherent from the previous model.

**Strengths:**

The main strength of the paper lies on a further theoretical analysis learning with concurrent verifiers.

**Weaknesses:**

One drawback is that the paper seems to be heavily based on the previous work by Niching et. al. The problem is that for a non-specialist in this area, it is a bit difficult to determine how distinct from the original paper the current submission is.

**Questions:**

Can you highlight more explicitly wha are the novel aspects of your paper? What are the parts that are essentially new, and what obstacles needed to be overcome to obtain the new results?

---

### Official Review · Reviewer_ibrb · 2023-11-01

**Soundness:** 2 fair
**Presentation:** 2 fair
**Contribution:** 2 fair
**Rating:** 3
**Confidence:** 2

**Summary:**

In this submission, the authors study the problem of learning with a concurrent verifier that verifies a prediction to be output or discarded. The submission is a continuation of the work "Generalization Analysis on Learning with a Concurrent Verifier" (Nishino et al 2022) that proves novel generalization bounds, generalizes the verification to requirement of multiple input-output pairs and they show that the learning problem remains a convex optimization problem when including a concurrent verifier.

**Strengths:**

- Provides tighter bounds
- Showing that the problem when learning with a concurrent verifier remains a convex optimization problem
- Extends the verification to multiple samples

**Weaknesses:**

- The paper is unfortunately very incremental
- The biggest weakness/concern is that the paper is not at all self-contained. A reviewer has to read the previous contribution; which disclosed the authors as multiple passages of the paper and figures are just copy and pasted from previous work. This makes the submission unfortunately not yet ready for publication at another venue.

**Questions:**

I thank the authors for their submission. My suggestion would be to have another writing pass over it to make it more self-contained and a clear contribution in itself without massive copies from prior work. Formally, copy and pasted figures from prior submissions would need to be cited to not disclose the authors, just to give one example of many.

---

### Official Review · Reviewer_H3HT · 2023-11-06

**Soundness:** 3 good
**Presentation:** 3 good
**Contribution:** 3 good
**Rating:** 6
**Confidence:** 3

**Summary:**

The paper is a theoretical study of the use of a concurrent verifier in machine learning models to ensure that the model's input-output pairs meet specific requirements. The paper proves several properties of using CV, such as the preservation of convexity and improvement on the generalisation error bound in certain situations. The paper also addresses some limitations of previous work, such as assumptions about the form of requirements and the complexity of evaluating them.

**Strengths:**

- The use of an additional module with a ML model to guarantee certain property seems to be a reasonable and realistic scenario, especially in safety-critical situations. Therefore, the topic of this paper is of interests to the ML community.
- The paper studies several aspects of using a CV, and considers more complex application scenarios (e.g., checking whether the property holds is imprecise) compared to a previous work. The contribution of the paper is novel as far as I'm concerned but I cannot judge its practical usefulness and relevance.
- The paper is relatively self-contained, making it not difficult to follow for someone like me that does not work in this domain.

**Weaknesses:**

- No experimental results are given.

**Questions:**

N/A

---

### Meta-Review · Area_Chair_TKSG · 2023-12-08

**Metareview:**

The paper studies the generalization properties of a network that is trained with a concurrent verifier that modifies its output to satisfy specific input-output constraints, and derives generalization bounds and other theoretical properties of classifiers trained in this manner. While the theoretical results are interesting, the reviewers brought up several issues with the quality of presentation and practicality of the results, which were unaddressed in the rebuttal. Hence, I recommend rejection but encourage the authors to submit a revised version to a future venue.

**Justification For Why Not Higher Score:**

Practicality of results and quality of presentation is lacking.

**Justification For Why Not Lower Score:**

N/A

---

### Decision · Program_Chairs · 2024-01-16

Reject